# Iranian livestock breeders' knowledge, attitude, practice, and behavioral determinants related to brucellosis prevention

Farhad Bahadori[1], Fazlollah Ghofranipour[1], Fatemeh Zarei[1], Saeideh Ghaffarifar[2]*, Reza Ziaei[3]

1 Department of Health Education and Health Promotion, Faculty of Medical Sciences, Tarbiat Modares University, Tehran, Iran, 2 Medical Education Research Center, Health Management and Safety Promotion Research Institute, Tabriz University of Medical Sciences, Tabriz, Iran, 3 Department of Health Sciences, Unit for Public Health Sciences, Mid Sweden University, Sundsvall, Sweden

* ghaffarifars@tbzmed.ac.ir

## Abstract

Brucellosis is a zoonotic disease that affects animals and humans. Its transmission to humans occurs through various routes such as consumption of infected animal products or unprotected close contact with secretions or different parts of live or dead infected animals. This study aims to report Iranian livestock breeders' awareness, attitude, practice, and behavioral determinants related to Brucellosis prevention. A cross-sectional study was conducted in 2019 among 450 livestock breeders in Beyraq, a suburb of Tabriz city. The Brucellosis Prevention Questionnaire (BPQ) was used to collect data. Statistical analysis performed using SPSS-23. The BPQ, consisting of 53 items, had acceptable psychometric properties (Content Validity Index = 0.90, Content Validity Ratio = 0.74, Impact Score = 4.30, Intra-class Correlation Coefficient = 0.885, Composite Reliability = 0.895, and Standard Error of Measurement = 5.448). The study surveyed 450 livestock breeders, with an average age of 51.68 ± 16.4 years.. Participants with a history of brucellosis reported that their last occurrence of the disease, on average, was 7.03 ± 5.83 years ago.Livestock breeders had moderate knowledge levels (mean score = 17.13) and positive attitudes (mean score = 3.86) towards Brucellosis prevention, but their practice level was relatively low (mean score = 15.9). Significant differences were observed in awareness (p-value <0.001), attitude (p-value = 0.03), and practice (p-value <0.001) scores between those who had undergone previous prevention measures compared to those who did not. Participants with a higher education level had higher awareness, attitude, and practice scores. An analysis of variance test (ANOVA) showed that job level had a significant effect on awareness (p-value <0.001) and practice (p-value <0.001) scores, with free jobs having higher scores than other jobs. Findings suggest that Iranian livestock breeders have insufficient knowledge about Brucellosis prevention despite positive attitudes and practices. To prevent the spread of Brucellosis, it is

**Data availability statement:** All relevant data are within the paper and its Supporting information files.

**Funding:** This study was supported by the Research Committee of Tarbiat Modares University and forms part of the Ph.D. dissertation of Dr.Farhad Bahadori at the same institution. The authors received no financial support for the research design, data collection, analysis, interpretation, and authorship. The funders had no role in study design, data collection and analysis, decision to publish, or preparation of the manuscript. The publication fee was covered by Mid Sweden University.

**Competing interests:** The authors have declared that no competing interests exist.

necessary to increase awareness and educate livestock breeders about preventive measures.

## Background

Brucellosis is a severe infectious disease caused by bacteria of the genus Brucella, which is capable of transmission between animals and humans. It is a major health problem in parts of Asia and the Middle East [1]. Approximately half of the countries in the Middle East, have a high prevalence of Brucella infections, posing significant public health and agricultural concerns [2,3]. Brucellosis can be transmitted from animals to humans through multiple pathways. One primary route of infection is the consumption of contaminated animal products, such as unpasteurized dairy products or undercooked meat from infected livestock. Additionally, direct contact with bodily secretions, tissues, or other biological materials from infected animals, whether living or dead. Furthermore, handling placentas or other reproductive tissues from livestock that have experienced abortion due to Brucella infection is another recognized mode of exposure, particularly for individuals working in agricultural or veterinary settings. [4,5]. According to statistics provided by the World Health Organization (WHO), approximately 500,000 individuals are infected with brucellosis annually worldwide [6,7]. Control and eradication programs play a crucial role in helping many countries reduce the prevalence of brucellosis in both humans and animals. However, in Iran, the infection remains a significant public health concern, with an estimated 15.4% of the population affected. This persistent prevalence continues to pose a major challenge for health organizations working to mitigate the disease's impact [7]. Several strategies are available for controlling brucellosis in livestock. These include vaccinating animals to prevent infection, properly disposing of contaminated products and biological materials from infected animals. Additionally, infected animals should be isolated during livestock trade to prevent disease transmission and should only be introduced into new stables after ensuring they do not pose a risk to healthy animals [8,9]. Among the previously mentioned control measures, livestock vaccination is regarded as the primary and most effective step in the eradication of brucellosis [10]. A lack of sufficient knowledge regarding the transmission of brucellosis, its health and economic consequences, preventive measures, and the risks associated with untreated infections hinders livestock breeders from effectively vaccinating their animals. Insufficient awareness and understanding of the disease contribute to lower vaccination rates, ultimately sustaining its prevalence among livestock and increasing the risk of transmission to humans [5,11].

In Iran, livestock can be vaccinated against brucellosis free of charge in all provinces and cities [11]. However, several factors contribute to low vaccination rates among livestock. Some of the key challenges associated with brucellosis vaccination include: Livestock breeders often lack sufficient knowledge about proper vaccination procedures. Additionally, vaccines may be administered at inappropriate times, reducing their effectiveness. Other challenges include improper vaccine storage, potential issues with vaccine efficacy, and inadequate quarantine conditions, all of

which contribute to the limited success of brucellosis vaccination programs.The risk of abortion due to brucellosis is not considered when vaccinating livestock [9].

Previous research on knowledge, attitudes, and practices (KAP) related to brucellosis among humans with frequent livestock contact in different areas where the disease was endemic has shown different and inconsistent findings. A study conducted in Kenya revealed a limited understanding among the population regarding the transmission of brucellosis from animals to humans [12]. A study conducted on small dairy farms in Kermanshah province, Iran, revealed that farmers had limited knowledge about brucellosis and frequently engaged in high-risk practices that could contribute to its spread [13].

Despite efforts by livestock breeders in Iran to control and eradicate brucellosis, the disease remains a significant public health concern. This challenge is particularly pronounced in rural areas, where livestock farming plays a crucial role in the local economy. A study in the Journal of Infection and Public Health found that in 2017, 16,482 cases of Brucellosis were reported in Iran [3,5]. High occurrence of Brucellosis not only puts people's health at risk but also has major negative effects on the livestock industry, such as lower productivity and the loss of important breeding animals [14]. Furthermore, Iranian livestock breeders face several challenges in preventing and controlling Brucellosis, including cultural barriers that may discourage vaccination and other prevention measures, lack of access to veterinary care, and limited financial resources.

These challenges underscore the need for targeted interventions and public health policies tailored to the specific needs and circumstances of Iranian livestock breeders. Additionally, raising awareness about the importance of brucellosis prevention and control strategies is essential for effectively reducing the disease's impact.

Although brucellosis has been the subject of several studies in Iran, much of the existing research has focused on national trends or broad public health aspects, often neglecting the specific conditions of smaller agricultural communities. This study distinguishes itself by focusing on Beyragh, a key hub for dairy production in the northwest of Iran. Despite its important role in the country's livestock sector, no previous research has investigated how breeders in Beyragh perceive and respond to the risks associated with brucellosis. By examining their knowledge, attitudes, and practices, this study provides valuable insight into the behavioral and contextual factors that may affect disease control in the region. The findings are expected to contribute to the design of more effective, locally adapted interventions and policies for brucellosis prevention, both in Beyragh and in similar rural areas across Iran.

## Methods and materials

### Study site and sampling

The present cross-sectional study was conducted in Beyraq. This village is situated on the northern slopes of the Sahand Mountains, in the suburbs of Tabriz, a metropolitan city in Iran. The village's climate, characterized by cold winters and mild summers, provides suitable conditions for raising sheep, goats, and cattle.The primary occupation of Beyraq's residents is animal breeding and dairy production, making the village a significant contributor to the country's cheese industry. More than 5,000 people live in Beyraq.

### Study design

The inclusion criteria for the study were those who were livestock farmers and lived in Beyraq. We prepared a sampling framework from a list of 2,122 livestock breeders living in Beyragh. Since livestock breeders delivered their dairy products to local dairy production mini-factories in the region, we created a complete list of livestock breeders from forty cheese production mini-factories in the village. We randomly selected 450 livestock breeders from the sampling framework using randomizer software (www.randomizer.org). The recruitment period started on 12-01-2019, and ended on 03-04-2019.

### Ethical consoderations

The faculty of medical sciences' ethical committee board at Tarbiat Modares University approved the present research (IRB number: IR.MODARES.REC.1397.001).The procedure was performed according to the relevant guidelines,

regulations, ethical standards of the responsible committee, which approved the study at Tarbiat Modares University, the Iranian Registry of Clinical Trials, and the Declaration of Helsinki (the revised version in 2000). The researchers confirm that all methods were performed following the relevant guidelines and regulations. Participants were informed about the present study's purpose before the investigation. Volunteer animal breeders participated in this study. The main researcher read the form for illiterate breeders; then breeders stamped the consent form. The research data are kept confidential. All participants had a unique code and could withdraw from the study at any time. Participants provided written informed consent and were ensured that their responses would remain confidential.

### Inclusivity in global research

Additional information regarding the ethical, cultural, and scientific considerations specific to inclusivity in global research is included in the Supporting Information (S1 Checklist).

### Data collection tool

We used the Brucellosis Prevention Questionnaire (BPQ) to collect the data. (S1 Questionnaire).

The BPQ has been developed and validated through an exploratory psychometric study as another part of our bigger study. The BPQ consists of 53 items and has acceptable psychometric properties (Content Validity Index = 0.90, Content Validity Ratio = 0.74, Impact Score = 4.30, Intra-class Correlation Coefficient = 0.885, Composite Reliability = 0.895 and standard error of Measurement = 5.448) [15,16]. Based on the results from exploratory factor analysis, the items of the BPQ had been loaded into awareness, attitude, and practice constructs. Awareness items had been categorized into three sub-constructs of "direct awareness", "indirect awareness" and "vaccine-oriented awareness". The sum of the scores obtained from these three categories was considered as the awareness score of the individuals [9].

Each question was explained to the livestock breeders, and their answers were recorded.

The questionnaire scores were calculated as the mean of the ratings for the items. However, if there was no response on two or more items, no score was computed. High scores indicate that participants have good understanding, while lower scores indicate a lack of some or all of these qualities. In the BPQ, an average awareness score of less than 0.33 indicates poor awareness, a score between 0.33 and 0.66 indicates average awareness, and above 0.66 indicates good awareness. An overall attitude score below 1.33 indicates a poor attitude, a score between 1.34 and 3.66 indicates an average attitude, and above 3.67 indicates a good attitude.

### Statistical analyses

SPSS software version 22.0 (IBM Corp., Armonk, NY, USA) was used for data analysis. Descriptive statistics were used to summarize participants' baseline characteristics, awareness, attitude, and practice scores. We conducted chi-square tests and independent t-tests to compare differences in awareness, attitude, and practice scores across various subgroups. A p-value of less than 0.05 was considered statistically significant. Additionally, multivariate linear regression analysis was performed to identify factors associated with awareness, attitude, and practice scores, adjusting for potential confounders such as age, gender, education level, and livestock production experience. The results of the regression analyses are presented as beta coefficients along with their corresponding 95% confidence intervals (CI).

## Results

### General characteristics of participants

All participants in the study were male, comprising 100% of the sample. The mean age of the participants was 51.68 ± 16.4 years. The mean household size was 6.47 ± 2.25, with a minimum of 2 and a maximum of 15. Participants who reported a previous history of Brucellosis in either themselves or their family members had experienced the disease

an average of 7.03±5.83 years ago. All participants were from the village, (100%) that 53 person (12.3%) were in contact with Sheep/Goat and 379 person (87.7%) were in contact with Cow. The majority of participants (76.2%, n = 329) were farmers. A total of 191 participants (44.2%) were illiterate, while 184 participants (42.6%) had attained elementary education. The income of 382 participants (88.4%) was less than their household expenses. Approximately 412 participants (95.4%) had a history of previous disease prevention measures, while only 176 participants (40.7%) reported no history of brucellosis in themselves or their families. A summary of qualitative demographic variables, presented as frequencies and percentages, along with participants' baseline characteristics, is provided in Table 1.

Out of the 450 participants, 432 completed the BPQ for assessing awareness scores (Response rate = %96). Among the respondents, 154 (35.6%) demonstrated poor knowledge, 240 (55.6%) had average knowledge, and 38 (8.8%) exhibited good knowledge regarding brucellosis prevention measures. All 450 participants completed the BPQ to assess attitude scores. The results indicated that a very small proportion (0.5%) exhibited a poor attitude toward brucellosis prevention measures, 96 participants (22.2%) demonstrated an average attitude, and the majority (77.3%) showed a good attitude.

## Practice scores

Practice scores were assessed using the same cut-off point method applied for attitude scores. Among the 450 participants, 432 completed the BPQ for the evaluation of practice scores. The results revealed that 55 participants (12.7%)

**Table 1. Baseline characteristics of the study participants* (n = 432).**

| Numerical Variables | | Mean± SD; (min-Max) |
|---|---|---|
| Age | | 51.68±16.40 (16-98) |
| The number of people in a household | | 6.47±2.25 (2 –15 ) |
| History of previous Brucellosis (years ago) | | 7.03±5.83 (1-30) |
| **Categorical Variables** | | **[N (%)]** |
| Gender | Male | 432 (%100) |
| | Female | 0 (%0) |
| Residential area | Rural | 432 (%100) |
| | Urban | 0 (%0) |
| Animal type | Sheep & Goat | 379 (%87.7) |
| | Cow | 53 (%12.3) |
| Another job besides livestock breeding | Self-employed | 16 (%3.7) |
| | Employee | 12 (%2.8) |
| | Farmer | 329 (%76.2) |
| | No more extra job | 75 (%17.3) |
| Educational level | Illiterate | 191 (%44.2) |
| | Elementary | 184 (%42.6) |
| | High school | 48 (%11.1) |
| | Graduate diploma | 9 (%2.1) |
| Household income | Less than household expenses | 382 (%88.4) |
| | Equal to household expenses | 50 (%11.6) |
| Previous participation in a Brucellosis prevention course | No | 412 (%95.4) |
| | yes | 20 (%4.6) |
| Previous history of Brucellosis among family members | No | 256 (%59.3) |
| | yes | 176 (%40.7) |

*Livestock breeders in Beiraqu.

exhibited poor performance regarding brucellosis prevention practices, 326 participants (75.5%) demonstrated average performance, and 51 participants (11.8%) showed good performance. The number and percentage of different categories of participants' average knowledge, attitude, practice are presented in Table 2.

Questions related to the awareness score have been presented into three categories of direct awareness score, indirect awareness score and Vaccine oriented behavior score, which is the sum of the scores obtained from these three categories as the awareness score of individuals. The frequency (percentage) of awareness or lack of awareness of participants in response to each awareness questions is presented in S1 Table.

The number and percentage of participants who responded to each of the attitude questions(divided by the different levels of the Likert scale: Agree, Strongly agree, Moderately agree, Disagree, and Strongly disagree) are shown in S2 Table. The results indicate that the majority of participants had a positive attitude towards Brucellosis prevention measures. For instance, more than 80% of participants agreed or strongly agreed that they would vaccinate their animals against Brucellosis.

The number and percentage of participants who selected the options of never, rarely, sometimes, often, and always in response to practice questions, are presented in S3 Table.

The findings suggest that there is potential for improvement in certain aspects of brucellosis prevention practices among livestock breeders. For instance, less than half of the participants consistently wore protective clothing while handling livestock, and approximately 30% reported always washing their hands after contact with animals.

**Association between awareness, attitude, and practice scores and demographic characteristics.** By using analysis of variance (ANOVA), a significant difference was observed between awareness score (p-value <0.001) and practice score, (p-value <0.001) with different levels of job class. Self-employed had higher scores for awareness, attitude, and practice compared to other jobs. Furthermore, a significant difference was observed between awareness score (p-value <0.001), attitude score (p-value = 0.03), and practice score (p-value <0.001) with different educational levels. The highest level of awareness was seen in individuals with a diploma education (M & SD, mean and standard deviation: 20.55±8.19), while the highest level of attitude score was seen in those with a high school education (M & SD: 4.53±37.62). Additionally, the highest practice score was reported by those with a high school education (M & SD: 19.44±7.26).

By using independent sample t-test, a significant difference was observed between awareness score (p-value <0.001), attitude score (p-value = 0.03), and practice score (p-value <0.001) with prior Brucellosis prevention. Individuals who did not have previous prevention had higher levels of awareness and attitude than those who had previous prevention, but those who had previous prevention had higher practice scores than those without previous prevention. There was also a

Table 2. The number and percentage of different categories of participants'* average knowledge, attitude, practice(n = 432).

| Constructs | Categories | Frequency | Percent |
| --- | --- | --- | --- |
| **Knowledge** | Weak | 154 | 35.6 |
| | Medium | 240 | 55.6 |
| | Well | 38 | 8.8 |
| **Attitudee** | Weak | 2 | 5 |
| | Medium | 96 | 22.2 |
| | Well | 334 | 77.3 |
| **Practice** | Weak | 55 | 12.7 |
| | Medium | 326 | 75.5 |
| | Well | 51 | 11.8 |

*Livestock breeders in Beiraqu.

significant difference between participants income and awareness score (p-value <0.001), attitude score (p-value = 0.05), and practice score (p-value = 0.02). Moreover, there was a significant difference between illness history in individuals or those around them with awareness score (p-value = 0.001), attitude score (p-value = 0.01), and practice score (p-value <0.001).

Finally, there was no significant difference between awareness score (p-value = 0.06) and attitude score (p-value = 0.09) with the livestock type. However, there was an almost significant difference in attitude score between cows and sheep/goats (p-value = 0.05).

**Correlations between awareness, attitude, and practice scores and demographic characteristics.** The results of Pearson correlation test showed that there is no significant relationship between age and awareness score (correlation = 0.05, p-value = 0.25), attitude score (correlation = 0.07, p-value = 0.14), and practice scores (correlation = 0.05, p-value = 0.29). Using Spearman correlation test, a relatively weak significant relationship was observed between awareness score (correlation = 0.21, p-value <0.001) and practice score (correlation = 0.20, p-value <0.001). Moreover, a relatively weak significant relationship was observed between awareness score (correlation = 0.15, p-value = 0.002) and practice score (correlation = 0.19, p-value <0.001) with income.Attitude of individuals (correlation = 0.10, p-value 0.03) and knowledge of individuals (correlation = 0.14, p-value 0.003) had a relatively weak positive relationship with prior Brucellosis prevention. Additionally, awareness score (correlation = 0.20, p-value <0.001) and practice score (correlation = 0.25, p-value <0.001) had a relatively weak positive relationship with illness history. Finally, there was a relatively weak significant relationship between practice score (correlation = 0.13, p-value = 0.008) and livestock type.

The level of knowledge, attitude and practice of participants among different categories of their baseline characteristics is presented in Table 3.

**Comparison of awareness, attitude, and practice scores by job and education level.** The mean awareness scores of individuals with self-employed jobs and farmers differed significantly at a level of less than (0.001). Furthermore, the mean awareness scores of unemployed individuals and farmers were significantly different at a level of less than 0.006. A significant difference was observed in the average awareness scores of illiterate individuals at a level of less than <0.001 and those with elementary education at a level of less than 0.003 compared to diploma education. However, there was no significant difference in attitude scores between individuals with diploma education and those with other levels of education. Moreover, no significant difference was observed between farmers and other jobs in attitude scores.

Regarding practice scores, a significant difference was observed between the mean practice scores of individuals with free jobs and unemployed individuals compared to farmers at levels of less than 0.001. Additionally, there was no significant difference in the mean practice scores across different educational levels compared to diploma education. The p-values related to the comparison of the average score of awareness score, attitude score, and practice score in different educational levels compared to a diploma, as well as different jobs compared to farmers, using the general linear model test are presented.

## Discussion

The findings of this cross-sectional study indicate that the majority of individuals in the study area possess knowledge regarding Brucellosis prevention, as more than two-thirds of the participants reported being aware of the disease. These findings are consistent with previous studies conducted in Pakistan, particularly among small-scale dairy farmers in the Punjab region. The studies reported that 61.3% and 87.3% of the farmers lacked knowledge about Brucellosis and its zoonotic transmission. However, the level of awareness about Brucellosis among individuals in Punjab, Pakistan, was higher compared to participants in the present study [17].

In Jordan, it was found that all interviewed livestock owners were aware of Brucellosis. Notably, 87% recognized the significant risk of infection from consuming unpasteurized milk, and 75% acknowledged similar risks from unpasteurized dairy products. However, awareness of other transmission routes was limited: only 19% identified fetal membranes as

**Table 3. The level of knowledge, attitude and practice of participants among different categories of their baseline characteristics.**

| Variables | | Kowledge | Attitude | Practice |
|---|---|---|---|---|
| **Another job besides livestock breeding** | Self-employed | 19.87±6.28 | 38.50±4.84 | 23.31±5.55 |
| | Employee | 13.50±4.54 | 35.83±7.96 | 14.83±4.41 |
| | Farmer | 13.49±5.40 | 35.23±7.08 | 16.08±6.75 |
| | No other extra job | 15.69±5.39 | 35.77±8.23 | 22.45±10.50 |
| **P-value*** | | <0.001 | 0.348 | <0.001 |
| **Educational level** | Illiterate | 12.95±4.94 | 35.88±6.70 | 15.65±6.83 |
| | Elementary | 14.31±5.60 | 34.57±8.12 | 18.87±8.78 |
| | High school | 16.73±5.63 | 37.62±4.53 | 19.44±7.26 |
| | Graduate diploma | 20.55±8.19 | 33.22±9.22 | 14.67±5.05 |
| **P-value*** | | <0.001 | 0.03 | <0.001 |
| **Household income** | Less than household expenses | 13.31±5.59 | 35.78±6.93 | 17.01±7.88 |
| | Equal to household expenses | 15.94±5.10 | 33.06±9.13 | 20.62±7.40 |
| **P-value**** | | 0.01 | 0.05 | 0.02 |
| **Previous participation in a Brucellosis prevention course** | No | 14.27±5.63 | 35.63±7.13 | 17.33±7.83 |
| | Yes | 10.85±252 | 32.00±8.98 | 19.20±9.28 |
| **P-value**** | | <0.001 | 0.03 | 0.30 |
| **Previous history of Brucellosis among family members** | No | 13.34±5.30 | 36.20±6.38 | 16.09±7.57 |
| | Yes | 15.22±577 | 34.39±8.28 | 19.36±800 |
| **P-value**** | | 0.001 | 0.01 | <0.001 |
| **Livestock type** | Sheep & Goat | 14.53±6.50 | 37.26±6.04 | 15.70±7.75 |
| | Cow | 14.05±5.43 | 35.21±7.38 | 17.66±7.90 |
| **P-value**** | | 0.61 | 0.05 | 0.09 |

* :One-way ANOVA test.

** : Independent T-test.

a risk, and just 13% were aware of the danger from contact with infected animals.On the other hand, a study conducted among urban farmers in Tajikistan found that they had comparatively lower levels of knowledge regarding Brucellosis and its prevention [18].

Research conducted in the Aseer Region of Southwestern Saudi Arabia reported that 90.0% of the 311 participants demonstrated a solid understanding of Brucellosis. Moreover, participants were considered to practice effective preventive behaviors, with 52.7% refraining from consuming meat from their own animals and 71.4% avoiding the slaughter of animals for food. This study found that the majority of participants exhibited a strong understanding, positive attitudes, and appropriate practices concerning Brucellosis. Gender and educational level were identified as key factors influencing the participants' knowledge, a finding that aligns with results from previous studies [19]. On the other hand, research conducted in regions of Ethiopia (Afar) and Somali found that 40.3% of livestock herds were affected by Brucellosis.

In the Afar region, the prevalence of Brucellosis among livestock ranged from 15.9% to 86.3%, whereas in the Somali region, it varied between 4% and 72.2%. Although the disease is widespread in both regions, many herders remained unaware of its existence and associated health risks. In contrast, a higher proportion of participants in the present study demonstrated awareness of Brucellosis, suggesting regional differences in disease recognition and knowledge among livestock breeders [19,20].

In Egypt, one study found that 67.4% of participants (70.0% of cases vs. 66.1% of controls) had heard of Brucellosis. Furthermore, the mean practice score related to animal husbandry, dairy processing, and consumption practices was significantly lower among the case group compared to the control group ($-12.7\pm18.1$ vs. $0.68\pm14.2$; $p<0.0001$),

this highlights a contrast with the findings of the present study regarding knowledge, attitudes, and practices related to Brucellosis (KAPs) [21]. Another study conducted in northwest Côte d'Ivoire found that veterinary interventions contributed to widening the knowledge gap on Brucellosis between men and women. While both male and female participants reported consuming raw milk, only men were found to have direct contact with animal waste without using any protective equipment. Studies suggest that improved control of Brucellosis depends on effectively communicating research findings to livestock-rearing communities, enabling them to adopt appropriate hygienic practices. The research indicated that strategies promoting adherence to hygiene practices and the use of personal protective equipment among both men and women were among the most effective measures for preventing Brucellosis [22].

On the other hand, research done in Khyber Pukhtun Khwa Province, Pakistan found that the people involved in the study had little knowledge about Brucellosis. The study looked at how much these people knew, how they felt, and how they acted concerning Brucellosis [23]. Moreover, when examining what doctors know about Brucellosis, it was discovered that only a small number of doctors had a good understanding of the disease. Most doctors did not possess enough knowledge about it [24].

## Conclusions

The findings of this study indicate that livestock breeders in Iran exhibit insufficient knowledge regarding Brucellosis prevention, despite demonstrating generally positive attitudes and practices. To effectively prevent the spread of Brucellosis, there is a critical need to enhance awareness and provide targeted education to livestock breeders about preventive measures.

Interestingly, the study suggests that individuals without prior exposure to Brucellosis prevention programs demonstrated higher levels of awareness and more favorable attitudes compared to those who had previously received preventive training. However, those with prior training exhibited better preventive practices, highlighting a potential gap between knowledge/attitude and actual behavior.

In the Beyraq region specifically, the study revealed low levels of knowledge, inadequate attitudes, and suboptimal practices related to Brucellosis prevention. Furthermore, there is currently no coordinated program in place for controlling the disease in the area. Given that knowledge, attitudes, and practices (KAP) serve as foundational elements for implementing effective health interventions, these findings underscore the urgent need for structured educational and public health initiatives.

This study emphasizes the importance of educating the public about Brucellosis, including its causes, transmission routes, symptoms, and preventive strategies as a key approach to reducing the risk of animal to human transmission. Public health education programs could play a vital role in enhancing community understanding and in controlling the spread of Brucellosis in the study area.

## Limitations and strengths of the study

The present staudy has some limitations. Firstly, In Iran, livestock breeding remains a male-dominated sector, particularly in rural villages, reflecting deep-rooted cultural, social, and religious norms. This gender disparity is further reinforced by the physically demanding nature of livestock management, which involves handling animals, transporting heavy feed, and performing labor-intensive tasks over extended hours in often harsh outdoor conditions.These demanding conditions align with the prevailing societal belief that physically heavy work is more appropriate for men. Only in some border regions, where nomadic communities migrate seasonally, do women participate in activities such as milking and feeding livestock. However, as previously noted, livestock breeding in Iran is predominantly male-dominated. At our study site, all livestock breeders were male. Therefore, the findings of this study can be generalized within the Iranian context, but may not be applicable to other countries. Secondly, Most of the livestock breeders who participated in the study had low levels of formal education or were illiterate. Therefore, the principal investigator was responsible for administering the questionnaires

through face-to-face interviews, verbally explaining each question to ensure participant understanding. To enhance accuracy, the investigator clarified responses and confirmed that the intended answers were correctly recorded. However, this approach may have introduced interviewer bias, as the investigator's presence and interpretation could have influenced the participants' responses. To minimize interviewer bias and ensure data accuracy, the principal investigator, who conducted all interviews, underwent prior training in standardized questionnaire administration and ethical communication with participants, particularly those with limited literacy. During data collection, care was taken to ask all questions neutrally and clarify responses without leading participants. After each interview, responses were reviewed immediately to verify consistency and completeness. Data entry was performed manually and cross-checked by a second reviewer to reduce the risk of transcription errors and ensure data reliability. Thirdly, data collection through self-report can be a strength; however, false-positive responses can limit its validity.

On the other hand, random selection of livestock breeders from all regions of Beyraq with different economic statuses was a strength of this study as it increases the generalizability of the results. The inclusion of educational levels was another strength of this study, as awareness of the correct methods of preventing Brucellosis and their performance in this regard may differ according to educational level. Overall, despite the limitations, this study's strengths increase its credibility and provide insight into the knowledge, attitudes, and practices towards Brucellosis prevention among male livestock breeders with lower levels of education in Beyraq.

## Implications

Based on the study findings, several actionable recommendations can be made for policymakers and veterinary institutions. First, there is a clear need to develop and implement structured and culturally appropriate educational programs targeting livestock breeders, particularly in rural and high-risk regions like Beyraq. These programs should focus on increasing awareness of brucellosis transmission routes, clinical symptoms, and effective prevention practices, including the importance of animal vaccination and safe handling of livestock products. Secondly, veterinary services should prioritize community outreach by deploying trained personnel to provide regular guidance, vaccinations, and monitoring. Policymakers should also consider allocating resources to establish a coordinated national or regional brucellosis control program, including mobile veterinary units, awareness campaigns through local media, and integration of brucellosis education into existing agricultural extension services. Strengthening collaboration between public health authorities and local veterinary departments can significantly enhance the implementation of these interventions and help reduce the public health burden of brucellosis.

## Supporting information

**S1 Checklist. Inclusivity in global research.**
(S1_Checklist.DOCX)

**S1 Questionnaire. Brucellosis Prevention Questionnaire (BPQ).**
(S1_Questionnaire.DOCX)

**S1 Table. The frequency (percentage) of awareness or lack of awareness of participants in response to each awareness questions.**
(S1_Table.DOCX)

**S2 Table. The number and percentage of participants who responded to each of the attitude questions.**
(S2_Table.DOCX)

**S3 Table. The number and percentage of participants' responses to each level of practice questions.**
(S3_Table.DOCX)

**S1 Data. Data set.**
(S1_Data.SAV)

## Acknowledgments

We would like to express our gratitude to all livestock breeders, who participated in this study, for dedication of their invaluable time and experience

## Author contributions

**Conceptualization:** Fazlollah Ghofranipour, Reza Ziaei.

**Data curation:** Fazlollah Ghofranipour, Fatemeh Zarei.

**Formal analysis:** Farhad Bahadori, Saeideh Ghaffarifar.

**Investigation:** Farhad Bahadori, Fazlollah Ghofranipour, Fatemeh Zarei, Reza Ziaei.

**Methodology:** Fazlollah Ghofranipour, Saeideh Ghaffarifar.

**Supervision:** Fazlollah Ghofranipour, Saeideh Ghaffarifar, Reza Ziaei.

**Validation:** Fazlollah Ghofranipour.

**Visualization:** Farhad Bahadori.

**Writing – original draft:** Farhad Bahadori.

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
