## [Decision Letter · Decision Letter 0]

17 Mar 2025

PGPH-D-24-03059

Iranian Livestock Breeders' Knowledge, Attitude, Practice, and Behavioral Determinants Related to Brucellosis Prevention

Dear Dr. Ghaffarifar,

Thank you for submitting your manuscript to PLOS Global Public Health. After careful consideration, we feel that it has merit but does not fully meet PLOS Global Public Health’s publication criteria as it currently stands. Therefore, we invite you to submit a revised version of the manuscript that addresses the points raised during the review process.

In accordance with the comments from the reviewers, please revise the following in the next version of the manuscript:

Provide a clear explanation for why the sample consisted exclusively of male respondents. Also address how this limitation may affect the findings and consider discussing the potential impact on the generalizability of the results.The introduction should better clarify the relevance of the study.In the discussion section, please expand on the broader implications of the findings, highlighting how they relate to existing research. Provide clear and actionable recommendations for policymakers, practitioners, or future researchers based on the study’s results. 

We look forward to receiving your revised manuscript.

Kind regards,

Delfina Fernandes Hlashwayo, Ph.D.

Academic Editor

Journal Requirements:

1. In the online submission form, you indicated that “The datasets used and/or analyzed during the current study are available from the corresponding author upon reasonable request.”.

a. In a public repository,

b. Within the manuscript itself, or

c. Uploaded as supplementary information.

Additional Editor Comments (if provided):

Reviewers' comments:

Reviewer's Responses to Questions

**Comments to the Author**

1. Does this manuscript meet PLOS Global Public Health’s publication criteria?

Reviewer #1: No

Reviewer #2: Yes

2. Has the statistical analysis been performed appropriately and rigorously?

Reviewer #1: No

Reviewer #2: Yes

3. Have the authors made all data underlying the findings in their manuscript fully available (please refer to the Data Availability Statement at the start of the manuscript PDF file)?

Reviewer #1: No

Reviewer #2: Yes

4. Is the manuscript presented in an intelligible fashion and written in standard English?

Reviewer #1: No

Reviewer #2: Yes

Reviewer #1: Thank you for your submission. While the study addresses an important public health issue, there are significant concerns regarding its contribution. The data, collected in 2019, is outdated, and no justification is provided for its current relevance. The study's focus on a single rural area with only male participants limits its generalizability, and the reliance on self-reported data introduces potential bias. Additionally, the discussion mainly reiterates the results without offering deeper insights or strong policy recommendations. To strengthen the manuscript, consider updating the data, expanding the sample for broader applicability, and providing clearer intervention strategies.

Reviewer #2: According to the questions above, the manuscript meet rigorously the PLOS GLOBAL PUBLIC HEALH publication criteria. the results study is according with the original research. All of the methods and statistical was described in sufficient detail to let it to understand easy and another researcher to reproduce the experiments. I was able to read the article and very calmly, the authors presented in the manuscript, without any restrictions, all the foundations of the results, in text and in the form of tables. Regarding language, I didn't note any specific errors. Its was presented in standard English and I appreciate it.

**Do you want your identity to be public for this peer review?** For information about this choice, including consent withdrawal, please see our Privacy Policy

Reviewer #1: **Yes: ** Ali

Reviewer #2: **Yes: ** Felisberto António Ussivane

---

## [Decision Letter · Decision Letter 1]

25 May 2025

PGPH-D-24-03059R1

Iranian Livestock Breeders' Knowledge, Attitude, Practice, and Behavioral Determinants Related to Brucellosis Prevention

Dear Dr. Ghaffarifar,

Thank you for submitting your manuscript to PLOS Global Public Health. After careful consideration, we feel that it has merit but does not fully meet PLOS Global Public Health’s publication criteria as it currently stands. Therefore, we invite you to submit a revised version of the manuscript that addresses the points raised during the review process.

We look forward to receiving your revised manuscript.

Kind regards,

Somayeh Hessam

Academic Editor

Journal Requirements:

Additional Editor Comments (if provided):

Reviewers' comments:

Reviewer's Responses to Questions

**Comments to the Author**

Reviewer #2: All comments have been addressed

Reviewer #3: All comments have been addressed

publication criteria?

Reviewer #2: Yes

Reviewer #3: Yes

3. Has the statistical analysis been performed appropriately and rigorously?

Reviewer #2: Yes

Reviewer #3: Yes

4. Have the authors made all data underlying the findings in their manuscript fully available (please refer to the Data Availability Statement at the start of the manuscript PDF file)?

Reviewer #2: Yes

Reviewer #3: Yes

5. Is the manuscript presented in an intelligible fashion and written in standard English?

Reviewer #2: Yes

Reviewer #3: Yes

Reviewer #2: It was a pleasure for me to be one of the reviewers of this manuscript. I have read it extensively and several times, and I can recommend it to everyone I know. The manuscript meets the PLOS GLOBAL PUBLIC HEALTH CRITERIA exactly. I fully understand the conclusion of the manuscript, as it is based on the data presented. I would like to see and read studies of this kind more often. It is very important that people understand the causes, modes of transmission and prevention of brucellosis, as it is a disease that poses a risk to public health.

Reviewer #3: Brucellosis is one of the most important public health challenges in low- and middle-income countries.The article is of good quality in terms of subject matter, method, and analysis, and can contribute to the field of livestock health and brucellosis control. However, some methodological and writing points require minor correction:

1- The article should better state its innovation more clearly in the introduction compared to previous studies in Iran

2-Due to the use of interviews with illiterate individuals, there is a possibility of interviewer bias; this should be highlighted in the limitations section.It would be better to provide more information on how to train interviewers and control data entry errors.

3-It would have been better to use multivariate regression analysis for all demographic variables to examine the adjusted effects more precisely.

4-Practical suggestions for policymakers and veterinary institutions could be clearer.

5-Some parts of the discussion are very long and should be structured in a more concise and targeted manner

6-Need for minor spelling revision in some English sentences (e.g. using correct tenses in verbs)

**Do you want your identity to be public for this peer review?** For information about this choice, including consent withdrawal, please see our Privacy Policy

Reviewer #2: **Yes: ** Felisberto António Ussivane

Reviewer #3: No

---

## [Decision Letter · Decision Letter 2]

15 Jul 2025

PGPH-D-24-03059R2

Iranian Livestock Breeders' Knowledge, Attitude, Practice, and Behavioral Determinants Related to Brucellosis Prevention

Dear Dr. Ghaffarifar,

Thank you for submitting your manuscript to PLOS Global Public Health. After careful consideration, we feel that it has merit but does not fully meet PLOS Global Public Health’s publication criteria as it currently stands. Therefore, we invite you to submit a revised version of the manuscript that addresses the points raised during the review process.

We look forward to receiving your revised manuscript.

Kind regards,

Somayeh Hessam

Academic Editor

Journal Requirements:

Additional Editor Comments (if provided):

Reviewers' comments:

Reviewer's Responses to Questions

**Comments to the Author**

Reviewer #3: All comments have been addressed

publication criteria?

Reviewer #3: Yes

3. Has the statistical analysis been performed appropriately and rigorously?

Reviewer #3: Yes

4. Have the authors made all data underlying the findings in their manuscript fully available (please refer to the Data Availability Statement at the start of the manuscript PDF file)?

Reviewer #3: Yes

5. Is the manuscript presented in an intelligible fashion and written in standard English?

Reviewer #3: Yes

Reviewer #3: The manuscript is mostly clear and logically structured. The abstract provides an accurate summary.

Minor revisions in the discussion section for clarity and conciseness are recommended

summarizing key findings from the supplementary tables in the main text for better accessibility

**Do you want your identity to be public for this peer review?** For information about this choice, including consent withdrawal, please see our Privacy Policy

Reviewer #3: No

---

## [Editor Report · Decision Letter 3]

8 Aug 2025

Iranian Livestock Breeders' Knowledge, Attitude, Practice, and Behavioral Determinants Related to Brucellosis Prevention

PGPH-D-24-03059R3

Dear Dr Ghaffarifar,

We are pleased to inform you that your manuscript 'Iranian Livestock Breeders' Knowledge, Attitude, Practice, and Behavioral Determinants Related to Brucellosis Prevention' has been provisionally accepted for publication in PLOS Global Public Health.

Best regards,

Somayeh Hessam

Academic Editor